# Lung Nodules Detection and Segmentation Using 3D Mask-RCNN

**Evi Kopelowitz**[1]                                                    evik@algotec.co.il

and **Guy Englehard**[1]                                        guye@algotec.co.il

[1] *Algotec LTD, a Carestream Co*

## Abstract

Accurate assessment of Lung nodules is a time consuming and error prone ingredient of the radiologist interpretation work. Automating 3D volume detection and segmentation can improve workflow as well as patient care. Previous works have focused either on detecting lung nodules from a full Computed Tomography (CT) scan or on segmenting them from a small Region Of Interest (ROI). We adapt the state of the art architecture for 2D object detection and segmentation, MaskRCNN, to handle 3D images and employ it to detect and segment lung nodules from CT scans. We report on competitive results for the lung nodule detection on LUNA16 data set. The added value of our method is that in addition to lung nodule detection, our framework produces 3D segmentations of the detected nodules.

**Keywords:** Lung Nodule detection, Convolutional Neural Network (CNN), 3D segmentation, Deep learning, Mask-RCNN

## 1. Introduction

Detection of lung nodules and accurate evaluation of their size are crucial for tracking cancer progression. Detecting the nodules is difficult since nodules vary greatly in shape and texture, and non-nodules such as vessels, fibrosis, diffusive diseases etc. have similar appearance to nodules. Once detected, nodules size is currently evaluated using Response Evaluation Criteria In Solid Tumors (RECIST). This measurement criteria relies on a linear measurement of the nodule along its largest axial slice. RECIST is shown to be inferior to a volumetric measurement (Welsh et al., 2012; Hayes et al., 2016). Nevertheless, it has become the standard of care since the time and effort required to manually delineate the 3D boundaries of nodules make such a workflow impractical for clinical applications. Therefore, an automated system that detects nodules and segments their 3D volumes can improve patient care by providing better information on disease progression, as well as reducing the time taken by radiologists to assess a lung CT study.

There is a large volume of work dedicated to detection of lung nodules on CT scans using 2D and 3D architectures. See for example (Ding et al., 2017; Jaeger et al., 2018; Setio et al., 2016). Similarly, previous works used CNNs to segment lung nodules from small ROIs (Nam et al., 2018; Feng et al., 2017; Qin et al., 2019; Wu et al., 2018).

(Jaeger et al., 2018) combined segmentation signals with object detection tasks to improve detection rates in various implementations of 2D and 3D networks. However, to date, no one reported on both object detection and segmentation results derived from one, end

to end, trainable network. We propose to adapt the MaskRCNN model (He et al., 2017), which achieves state of the art results on various 2D detection and segmentation tasks, to detect and segment lung nodules on 3D CT scans.

## 2. Methods

**Architecture.** MaskRCNN is a 2-stage object detector (Region Proposal Network (RPN) followed by Region based Convolutional Neural Network (RCNN) and a semantic segmentation model (MASK)). We modify the 2D implementation of MaskRCNN (Abdulla, 2017) to handle 3D images and to account for small object detection. Details regarding the full implementation of the model can be found in Appendix A and (Kopelowitz, 2019).

**Training.** 3DMaskRCNN is fully trainable end to end. Nonetheless, convergence is faster when training the backbone and RPN together first, and then training only the second stage heads. Focal loss (Lin et al., 2017) and Intersection Over Union (IOU) loss improve results in the class and MASK heads respectively. Training both segmentation and detection tasks simultaneously improves detection rate, similar to (Jaeger et al., 2018). We use dropout and heavy augmentation during training to avoid overfitting. We perform 10-fold cross validation.

**Inference.** We scan each image with overlapping sliding windows. Overlapping boxes are filtered using Non Max Suppression (NMS). To reduce False Positive (FP)s, we keep only boxes with a segmentation mask volume $> 0$. We use our in house lung mask CAD to remove nodules detected outside of the lungs.

## 3. Experiments and Results

We tested our model on the LUNA16 challenge, taken from the LIDC/IDRI database (III et al., 2015), which includes 888 CT scans. The reference standard of the challenge consists of all nodules $>= 3$ mm accepted by at least 3 out of 4 radiologists (Armato et al., 2011).

Detection evaluation is performed using the Competition Performance Metric (CPM), defined as the average sensitivity at 7 predefined FP rates: 1/8, 1/4, 1/2, 1, 2, 4, 8. Radiologists performance was evaluated in two cases: (1) Including only nodules $> 3$ mm and (2) including all nodules. Sensitivity and FPs were averaged over the 4 individual performances with respect to the other three. The results are summarized in Table 1 together with state of the art method (Ding et al., 2017) and ZNET, the winner of the LUNA16 challenge.

Note that our network achieves CPM of 0.826 with a single inference step, beating the winning result of the challenge. Improved detection results (score of 0.86) were obtained by performing a second FP reduction step, in which the model is fed with centered patches around proposed nodules. Although (Ding et al., 2017) reports on 15 Candidates per scan, 3DMaskRCNN achieves the highest sensitivity at 7-8 FPs per scan since the average number of True Positive (TP)s per scan $< 2$.

Nodule segmentation results are shown in Appendix B, Figure 2 demonstrating both small and large nodules as well as solid and ground-glass nodules.

Table 1: Comparison of Detection results

| Model | Sensitivity | FPs/scan | CPM |
|---|---|---|---|
| Radiologists ($> 3mm$) | 0.75 | 1 | NA |
| Radiologists (all) | 0.85 | 5 | NA |
| 3DMaskRCNN (ours) FP reduction | 0.936 | 7 | 0.86 |
| 3DMaskRCNN (ours) | 0.932 | 8 | 0.826 |
| 2DRcnn + 3DCNN (Ding et al., 2017) | 0.946 | < 15 | 0.891 |
| ZNET (Setio et al., 2016) | NA | NA | 0.811 |

Segmentation overlap is measured with the Dice Similarity Coefficient (DSC). Table 2 lists these results. Our results are comparable with radiologists' agreement (as calculated from their individual segmentations). Comparing to competing methods is difficult as other papers show segmentation results for predefined ROIs whereas our results are over the full 3D scan. Note, that the test set used in (Nam et al., 2018) contains only 113 nodules whereas ours has over 1000 nodules.

Table 2: Comparison of Segmentation results

| Model | DSC |
|---|---|
| 3DMaskRCNN (ours) | $70 \pm 10$ |
| Radiologists | $76 \pm 16$ |
| CNN on diameter(Nam et al., 2018) | $79 \pm 19$ |
| PN-SAMP-S1(Wu et al., 2018) | $74 \pm 3.57$ |

We evaluate the correlation between predicted segmentations volume and Ground Truth (GT) and found a strong correlation of 0.96, indicating that volumes are indeed a reliable measurement for size. The accuracy of the boundaries is assessed with Hausdorff Distance (HD) and is $2.49mm \pm 2.05mm$. The fact that the standard deviation is of the same size of the HD suggests that this measurement may be irrelevant for small object segmentation.

## 4. Conclusions

We show that 3DMaskRCNN can achieve competitive results for both detection and segmentation tasks. We demonstrate strong correlation between predicted volumes and GT and suggest that nodules volume evaluated and predicted by our model is a reliable measure of nodules size and may replace manual segmentation. As most of the FPs our model detects seem like genuine nodules, see Figure 3 in Appendix B, we believe that continued training/testing in order to further improve the CPM will inadvertently cause overfitting to the LUNA dataset and hurt generalization on unseen studies as we are continually testing on the same test data. A known problem with the LUNA dataset is that the GT is the intersection of 3-4 radiologists' detections, resulting in a very limited and strict dataset, very unlike a typical output of a single radiologist (Armato et al., 2011). We plan on training 3DMaskRCNN on a wider dataset in order to generalize our results.

## Acknowledgments

The authors thank Amir Yaacobi for his contribution in the implementation of the 3DMaskR-CNN architecture as well as Ohad Silbert, Hadar Porat and Zahi Peleg for fruitful discussions. The authors acknowledge the National Cancer Institute and the Foundation for the National Institutes of Health, and their critical role in the creation of the free publicly available LIDC/IDRI Database used in this study.

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

## Appendix A. 3DMaskRCNN Model Architecture

The 3DMaskRCNN is composed of four parts: backbone, RPN, RCNN for classification and bounding box regression and another CNN for pixel segmentation of objects, which we refer to as MASK.

**Input.** Images are rescaled to a resolution of 0.5 mm per pixel, and cropped into patches of size $128^3$). To reduce FPs, we concatenate positive and negative patches (Pan, 2018). Positive patches contain at least one nodule. The concatenated patches are normalized to have zero mean and unit variance.

**A.1 Backbone.** We implement the Inception Resnet v2 model (Szegedy et al., 2016) in 3D. In the reduction blocks we replace pooling layers with kernel dilation of 2 and 3 (known as Atrous kernels, (Chen et al., 2016)), thus achieving a wider field of view while maintaining full resolution. The output of the network is a feature map comprised of the output of the first and last reduction steps.

**RPN.** The RPN model operates on the feature map outputs of the backbone. Each pixel in the feature map is scanned with two anchors (of sizes 16 and 64 with a ratio of 1). We apply a bounding box regression and classification on every anchor. RPN proposals with scores greater than 0.1 are passed to the ROI Align layer. If no proposal scores higher than 0.1 are found, the 10 anchors with highest scores are passed. The low thershold was chosen to reduce False Negative (FN)s. Proposals and anchors are considered positive(negative) if their IOU with a ground truth box is greater(lower) than 0.5(0.1). The ROI Align layer crops the proposals from the feature maps and rescales them to a fixed size.

**RCNN and MASK.** RCNN receives the aligned proposals and applies several convolution layers to predict final object classification and bounding box regression. The MASK head of the network receives dilated ROIs (5 mm on each edge) in order to get a wider view of the nodule. Then, several convolution and deconvolution layers are applied to predict pixel level nodule segmentation.

During training we add dropout layers in the RPN and RCNN class heads. Training parameters are: $lr = 0.01$ (reduced by half on plateau) for the backbone and RPN. $lr = 0.001$ (reduced by half on plateau) for the RCNN and Mask heads. Momentum is set to 0.9 throughout training, and Stochastic Gradient Decent (SGD) optimizer is used. Each part is trained for 100 epochs. We perform 10-fold cross validation as required by the challenge in order to predict on the full data set. We implement our model using Keras Tensorflow. Please refer to (Abdulla, 2017) to find the full detailed description of each layer in the MaskRCNN.

## Appendix B. Figures

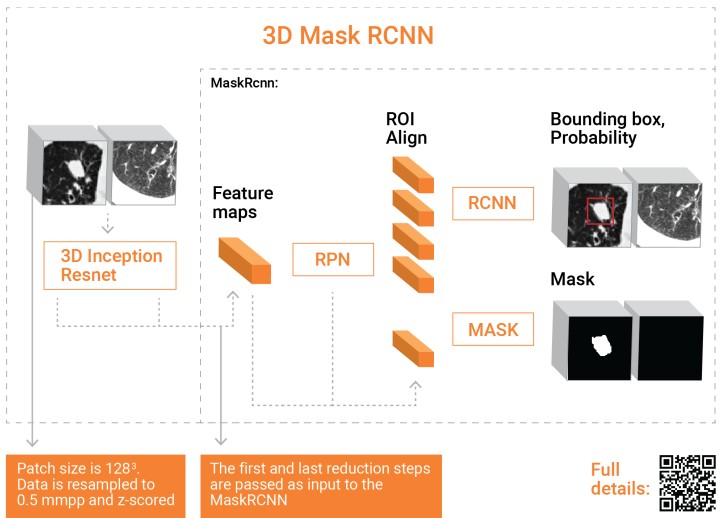

Figure 1: Diagram of the 3DMaskRCNN model

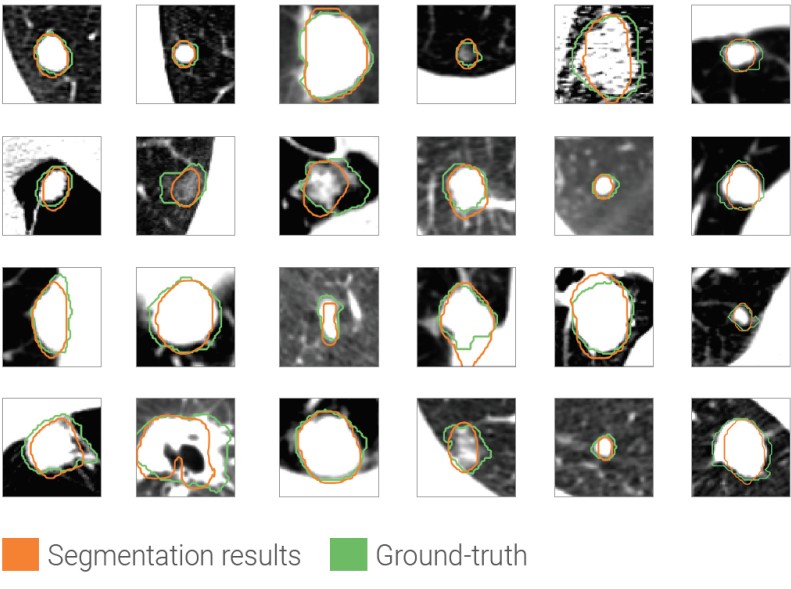

Figure 2: Examples of Nodule segmentation with 3DMaskRCNN. Box size is $3cm^2$

True positives

False positives

Figure 3: Examples of nodules detected by 3DMaskRCNN

