# OpenReview forum: "Lung Nodules Detection and Segmentation Using 3D Mask-RCNN"
_MIDL.io/2019/Conference/Abstract — MIDL Abstract 2019_

### Official Review · AnonReviewer1 · 2019-04-28
**improve originality (beyond 2d-3d adaptation), assess quantitatively differences of segmented shapes (hausdorff, possibly more adequate), highlight advantages of the method**

**Rating:** 2
**Confidence:** 3

**Review:**

Lung Nodules Detection an dSegmentation using 3D Mask-RCNN

The paper uses MaskRCNN to jointly detect and segment lung nodules in CT images. The method adapts MaskRCNN from 2D to 3D. Results indicates a lower scores than competing methods.
The submission may benefit from a clear statement on originality beyond a 2d-3d adaptation (what were the real methodological challenges, or is it purely a clinical contribution?). The results also look to indicate large differences in shapes between ground truth and predicted segmentations. The quality of the boundaries is typically assessment in segmentation papers, for instance, using the Hausdorff distance. If a lower score is shown (table 2, 70% vs competing 74-79%), the clear advantage should be highlighted. In other words, why should this method be used over the others? Is it accuracy (not shown here), is it speed (maybe), is it computational resources (maybe), what is the advantage?

---

### Official Review · AnonReviewer2 · 2019-05-01
**3D MaskRCNN for lung nodule segmentation**

**Rating:** 3
**Confidence:** 2

**Review:**

The abstract presents a 3D MaskRCNN model for joint lung nodule detection & segmentation from CT scans.

The closest related work might be by Liu et al. 2018, with 2D Mask RCNN for lung nodule segmentation (and detection).

There is a lot of work on nodule detection; however lung nodule segmentation is still a challenging task, and it is a sensible application of the MaskRCNN framework (with the additional 3D challenge).

The abstract is generally clear with adequate details given the format. The authors provide comparison to some competing methods in the literature.

---

### Decision · Program_Chairs · 2019-05-06
**Acceptance Decision**

Accept